# Reinforcement Learning for the Beginning of Starcraft II Game

**Yukang Chen**
1155154501
Department of Computer Science
The Chinese University of Hong Kong
yukangchen@cse.cuhk.edu.hk

**Chu Ruihang**
1155156766
Department of Computer Science
The Chinese University of Hong Kong
rhchu@cse.cuhk.edu.hk

## Abstract

Starcraft II is a popular real-time strategy game that is welcomed by many young people. This game is really complicated to play due to the long game timeline, various society/buildings/units, a large number of actions or selections, different constraints (*e.g.*, population limit), and the partially observed environments. In this project, we plan to develop a reinforcement learning model for the beginning of Starcraft II game, instead of the full-length game. The beginning of the game is essential for the further economy, population increase, and technology development. Our project is based on the SC2LE (StarCraft II Learning Environment) platform. We build a feasible pipeline for training reinforcement learning models and design random, scripted, and our actor-critic based agents. Experiments show that our actor-critic based agents can learn valuable knowledge in this task. The video has been publicly available. [1]

## 1   Introduction

In recent years, reinforcement learning (RL) has been widely used in computer vision, natural language processing, robotics (8), and so on. In terms of games, RL achieves remarkable success in Atari games (11) and the Go game (17). Based on these achievements, researchers gradually pay attention to the more complicated games, *e.g.* the real-time strategy (RTS) video games Starcraft II. In RTS games, players make their actions simultaneously, instead of acting in turns. This kind of games are more difficult to play well. It costs common teenagers several years to become a master in Starcraft II. Therefore, it is also uneasy to train an agent to play this game. We list the difficulties of this problem as the following:

- **Large action space**: If the agent uses the point-and-click interface just like human players, there would be hundreds of millions of possible actions (22) in each frame, for the high game resolution. In addition, with the game time going, players will possess different units, buildings, and technologies, which involves additional unique actions.

- **Indistinct reward**: The clearest reward is the results of the game (wining or losing). However, as each game time involves thousands of frames and actions in the sequence, it is too sparse to use the final reward as instruction. What's more, apart from the micro-level actions, macro-level plannings are important for the results of the game. It is a hard job to design an approximate reward function that is neither too sparse nor shortsighted.

- **Partially observation**: The environment is partially observed for two reasons. First, the map camera is locally posed. Players need to move the view to get more information. Second, there is a "fog-of-war" to cover an unexplored area on the map. Players need to assign units to find out the positions and activities of their enemies.

---

[1] https://drive.google.com/file/d/11S6t3rNKjM1CJEkFSN1lkZiyLTdqHhzQ/view?usp=sharing

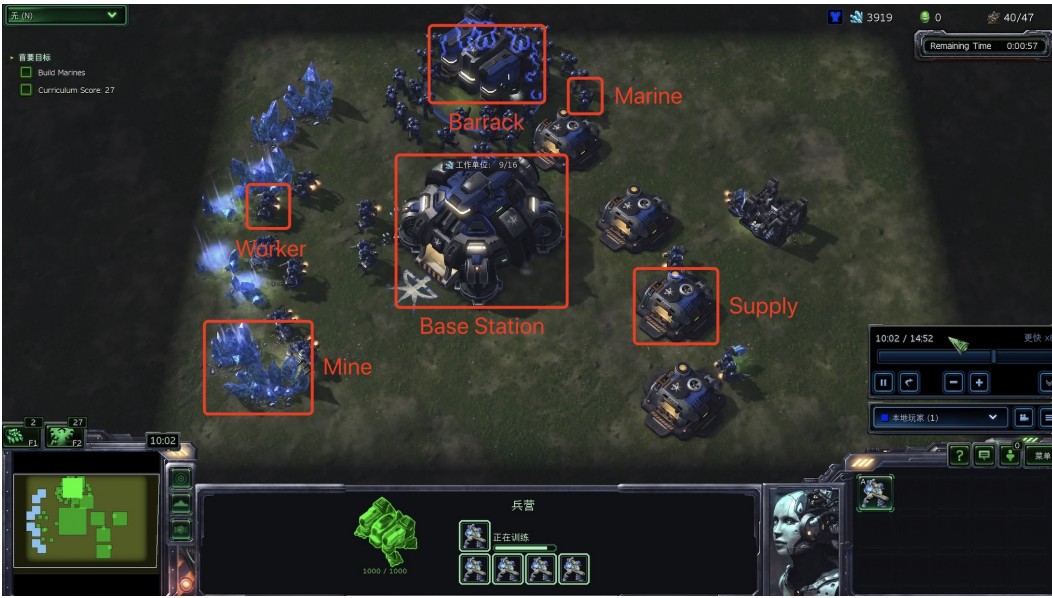

Figure 1: Illustration of units and buildings involved in the begining of this game.

In this project, we focus on the game beginning, which is essential for the full game and is also a complicated task. The beginning of the game is to set up the necessary units and buildings in a reasonable order and do their right jobs. It has less difficulty compared to the full game but is an essential task. Figure 1 1 shows an example layout of the beginning of the games with units and buildings noted. We design actor-critic agents to learn how to collect the economy and build marines and compare them to random and scripted agents. Our experiments are based on SC2LE, a StarScraft II Learning Environment platform, which is developed by the collaboration of DeepMind and Blizzard. It provides rich environments and interfaces for our agents to interact with StarCraft II. In Section 2, we introduce related works. In Section 3, we introduce our baseline agents and our RL based agents. We show the experiments in details in Section 4.2 and make a conclusion in Section 5.

## 2 Related Work

Reinforcement learning (RL) is studied to be suitable for playing real-time strategy games, achieving great advancements in classic board games (1; 17) and video games (10; 14; 5). Containing much theoretic and domain complexities, RL's playing Starcraft II gains increasing research attention. To facilitate it, DeepMind developed SC2LE (22), an RL environment for StarCraft II. It standardizes critical RL components such as the observation, action, and reward for Starcraft II domain, based on which various approaches could be fairly verified.

With a long-term measurable goal of beating human players, recent works (19; 16; 1) explores RL algorithms and architectures to improve win rates, with hand-crafted sub-systems (2), high-level actions(18), rule-based systems (3) and so on. Many works (18; 7; 13) also described the hierarchical reinforcement learning paradigm (6; 20; 12). Specifically, they generally operate on multi-level abstraction, each handled by learned strategies and controllers. Owing to its hierarchical architecture, the long-horizon task is decomposed into a series of sub-policies, along with it the huge action could be substantially reduced. As a milestone, AlphaStar (21) exploited a model-free learning method that uses data from human and agent game, as well as avoiding the imperfect models, which is typical of search-based methods. Finally, it reached the grandmaster level in this game.

Due to the severe difficulty of the full task of Starcraft II, *i.e.,* win the game, Blizzard and DeepMind designed seven minigames (22; 4) as well. Each can be regarded as a sub-task and focuses on different perspectives of the game, such as building a base and collecting resources. BuildMarines, as one of mini-games, encourages the agent to build as many Marines as possible within a fixed amount of time. Some earlier works (15; 22) searched low-level actions brutally with many meaningless attempts,

Actions: select_rect(P$_1$, P$_2$)     build_supply(P$_3$)

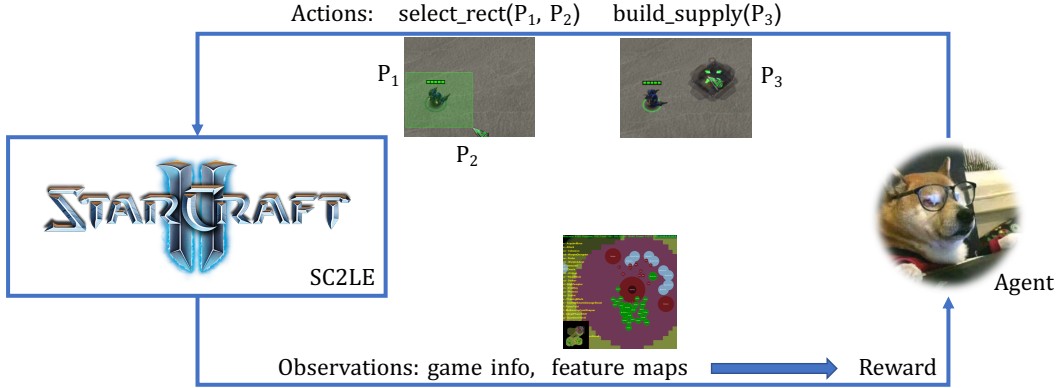

Figure 2: The interaction of the Starcraft II learning environment (SC2LE) platform and the agent.

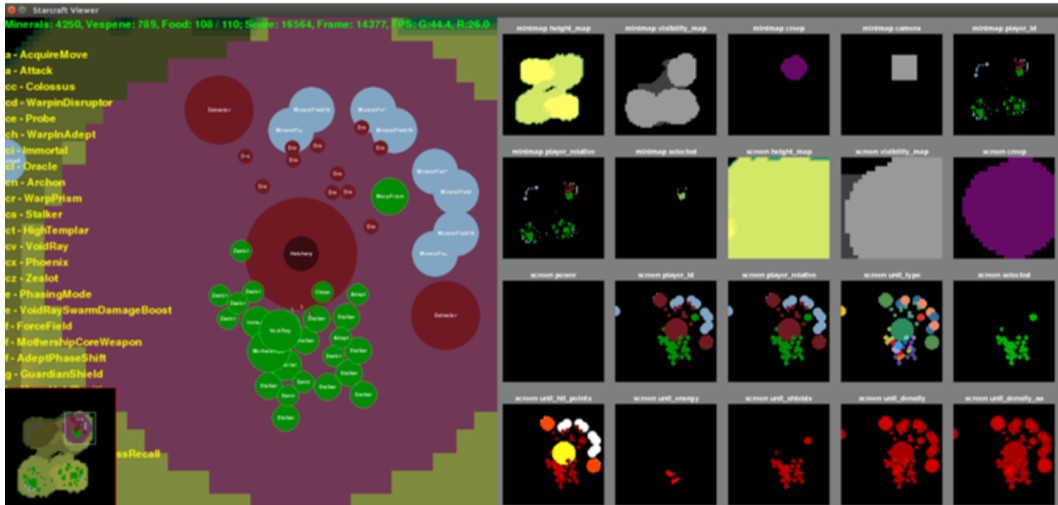

Figure 3: Processed input feature map in SC2LE platform.

thereby causing unsatisfactory results. By contrast, Liu. *et.al* (9) tackled this problem by splitting it several into basic skills. Then the DQN reinforcement learning algorithm and a ruled-based method were proposed to schedule and execute these skills, respectively. In this proposal, we aim to improve the performance from the different aspects, *i.e.,* model-free agent construction.

## 3   Method

### 3.1   Problem Formulation

In Figure. 2, we illustrate the interaction process of the SC2LE platform and the agent. In the following, we introduce the details of each parts in this system and the approach we adopt.

- **Environment**: The environment is the Starcraft II API in SC2LE. It receives the actions from the player or agent, interacts with the game, and returns the corresponding observations.
- **Action**: The agent actions in this platform directly follow the form of human actions. For example, to control a unit, it first selects the action 'select_rect', and then picks two points (P$_1$, P$_2$) of a rectangle, which should cover the unit. To further order the unit to do something, *e.g.*, to build a supply, it first selects this action 'build_supply' and then chooses the point P$_3$.
- **Observations** The observations involve two aspects, the game information, and feature maps. The game information includes resources, available actions, building queues, and

---

**Algorithm 1:** Random Agent

---

Initialise $s$;
**for** *each step* **do**
  Randomly sample $a$ from $A_s$, available action set in $s$;
  Get transition $s', a, s, r$;
  $s \leftarrow s'$;
**end**

---

---

**Algorithm 2:** Scripted Agent

---

BUILD-SUPPLY $\leftarrow$ False;
BUILD-BARRACK $\leftarrow$ False;
Initialise $s$;
**for** *each step* **do**
  Command all available workers to mine;
  Do the following judgements based on $s$;
  **if** *not BUILD-SUPPLY* **then**
    Choose a worker to build a supply;
    BUILD-SUPPLY $\leftarrow$ True;
  **end**
  **else if** *not BUILD-BARRACK* **then**
    Choose a worker to build a barracks;
    BUILD-BARRACK $\leftarrow$ True;
  **end**
  **else**
    Command the barrack to build marines
  **end**
  **if** *Population meet the limit* **then**
    BUILD-SUPPLY $\leftarrow$ False;
  **end**
  Summary the current observation as $s'$;
  $s \leftarrow s'$;
**end**

---

other visible information. Feature maps are the processed segmentation maps of screen maps. This kind of feature map is easier for the agent to learn than the original screen. We have many different kinds of feature maps for use as shown in Figure 3.

## 3.2 Baseline Models

Before conducting our training process, we set up two baseline models, random agent and scripted agent. We introduce them in detail in this section.

**Random Agent**  Random agent is a trivial naive model, which randomly samples action from the available action choices in each frame. Its outputs are meaningless. Thus, it gets unsatisfied results. We mark the random agent as a brief algorithm in Algorithm 1.

**Scripted Agent**  Actually, we have enough experience and know the rules about this RTS game. Thus, in addition to the scripted agent, we also include a well-designed scripted agent as a baseline. It totally follows a series of if-else rules and case judgment. It checks the current situation and responds to an appropriate action from the current valid action set. Specially, because the purpose of this game is to build marines, which is trained from the barrack, this agent checks if the barrack has been built. If not, a worker will be selected and build a barrack. At the same time, other workers are mining (collecting money). After that, it builds marines continuously until it counters the population limit. Otherwise, a worker will be selected to build a supply, which can enhance the population limit. This

**Algorithm 3:** actor-critic Algorithm

---

Initialise $s, \theta$;
Sample $a \sim \pi_\theta$;
**for** *each step* **do**
  Perform $a_t$ according to policy $\pi(a_t|s_t; \theta)$;
  Receive reward $r = R_s^a$; sample transition $s' \sim P_s^a$;
  Sample action $a' \sim \pi_\theta(s', a')$;
  $\delta = r + \gamma Q_w(s', a') - Q_w(s, a)$;
  $\theta = \theta + \alpha \nabla_\theta log \pi_\theta(s, a) Q_w(s, a)$;
  $w \leftarrow w + \beta \delta \phi(s, a)$;
  $a \leftarrow a', s \leftarrow s'$;
**end**

---

process is repeated until the end. We formulate this process in Algorithm 2. Noted that we do not include very detailed actions in the algorithm, like how to select a worker, for a clear illustration.

### 3.3 Training Strategy

We design our training strategies based on actor-critic () and Asynchronous Advantage actor-critic (A3C) () algorithms. We first introduce the reward function. Then, we introduce the detailed design as follows.

**Reward**: For the game beginning task with regard to building Marines, here we directly correspond the reward value with the marine number, *i.e.,* +1 if increasing a marine. It is different from the full game where the reward function is just the winning or losing (+1/-1).

**actor-critic**  To deal with the indistinct reward mentioned in Section 1, we adopt actor-critic architecture that directly parameterizes the policy $\pi_\theta(a|s)$ and updates the parameters $\theta$ by performing gradient ascent on $E[R_t]$. In this approach, the critic plays a role as evaluating the reward of actions, guiding the policy function used to generate actions, denoted as

$$Q_w(s, a) \approx Q^{\pi_\theta}(s, a) \tag{1}$$

where $Q(s, a)$ is the state-action value function, the critic and actor are responsible for updating the actor parameter $w$ and policy parameter $\theta$. In our project, the state $s$ is the real-time game status as well as segmented screen map, action can be represented by vectors where each dimension is an action category. The pseudo-code is shown in Algorithm 3.

Under this strategy, the $w$ and $\theta$ are updated by steps until convergence.

**Asynchronous Advantage actor-critic**  Asynchronous Advantage actor-critic (A3C) () is an updated version of actor-critic algorithm. It accumulates the training process of the traditional actor-critic algorithm in an asynchronous way. In other words, the algorithm is running on multi threads in parallel. At the end of each epoch, the gradient is accumulated from multi-threads and used to update the global parameter. Then the thread-specific parameters are synchronized with the global one. We illustrate this algorithm in Algorithm 4.

## 4 Experiments

### 4.1 Implmentation Details

We train our models and compare them with two agents: RandomAgent and ScriptAgent. RandomAgent randomly chooses an action at every time step during the game, while ScriptAgent is well-designed with hand-craft playing strategies. Specifically, We control ScriptAgent to focus on quickly mining resources and building Marines, without many irrelevant procedures. When the Marines number goes up to an upper limit, ScriptAgent turns to build another Supply Station so as to accommodate more Marines. As for our agent, we train it on a MacBook Pro equipped with a 4GB Radeon Pro 555X GPU. For implementation details, we set the feature map resolution of both

**Algorithm 4:** Asynchronous Advantage actor-critic Algorithm

---

Initialize global shared parameter $\theta$; Assume thread-specific parameter $\theta'$;
Reset $d\theta \leftarrow 0$;
Initialize $s, \theta$;
Sample $a \sim \pi_\theta$;
**for** *each step* **do**
    Perform $a_t$ according to policy $\pi(a_t|s_t; \theta)$;
    Receive reward $r = R_s^a$; sample transition $s' \sim P_s^a$;
    Sample action $a' \sim \pi_\theta(s', a')$;
    Accumulate gradients wrt $\theta'$ $d\theta$;
    $\delta = r + \gamma Q_w(s', a') - Q_w(s, a)$;
    Perform asynchronous update of $\theta$ usirng $d\theta$.;
    $\theta = \theta + \alpha \nabla_\theta log\pi_\theta(s, a)Q_w(s, a)$;
    $w \leftarrow w + \beta\delta\phi(s, a)$;
    $a \leftarrow a', s \leftarrow s'$;
**end**

---

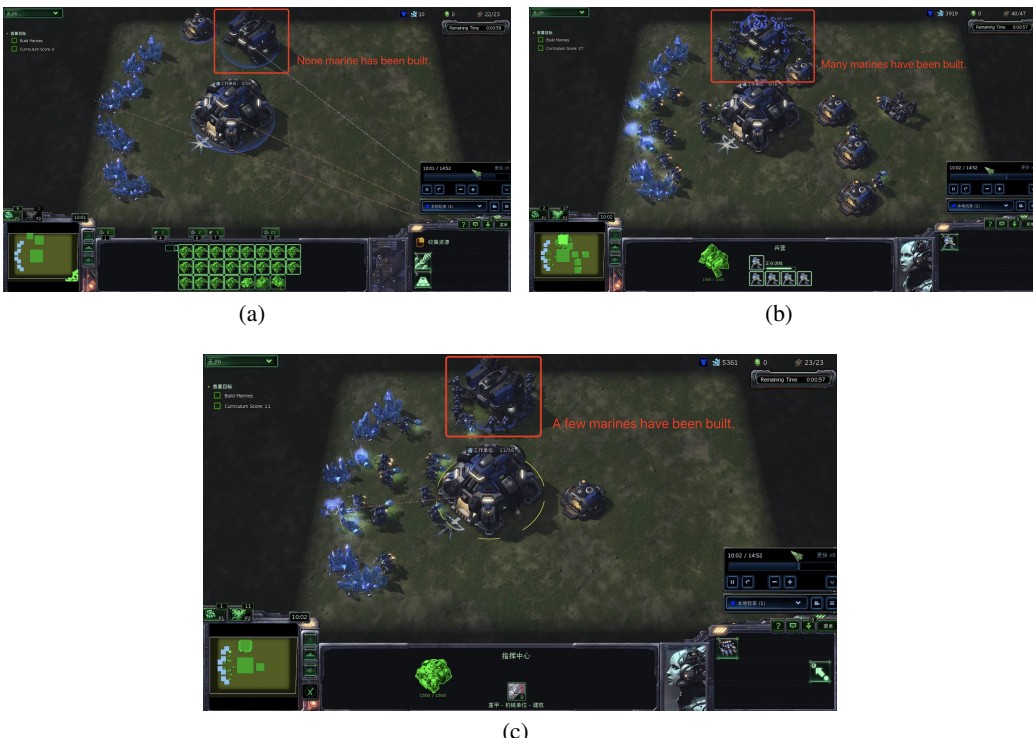

Figure 4: Comparison of (a) RandomAgent, (b) ScriptAgent, (c) Our Agent. (a) RandomAgent randomly sample actions at each frame and fail to build any marine to the end; (b) ScriptAgent builds marines in a normal way; (c) Our trained actor-critic agent has learned to build marines but is stuck by the population limit.

screen and minimap as 64×64. The feature extracting neural network is composed of 4 convolution layers and 6 fully connected layers. The maximum steps in one iteration is set as 2000 and the total iterations is 340. The learning rate is 5e-4, which is a carefully tuned value. The discount ratio 0.99.

## 4.2 Experimental Results

We evaluate the agent performance by the curriculum score, *i.e.,* the Marine number within a certain time. We show the experimental results by screenshots as shown in Fig 4. It can be observed that

| Agent | Curriculum score |
|---|---|
| RandomAgent | 0 |
| ScriptAgent | 27 |
| actor-critic Agent | 11 |
| A3C Agent | 11 |

Table 1: Curriculum score comparison between different agents.

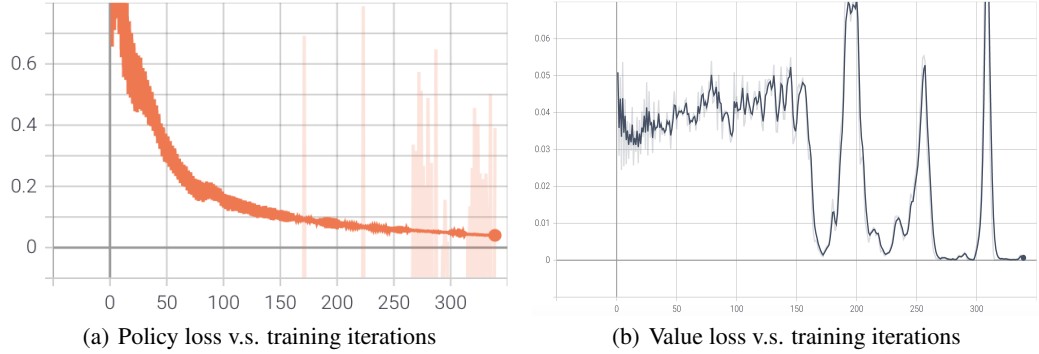

(a) Policy loss v.s. training iterations        (b) Value loss v.s. training iterations

Figure 5: (a) Policy loss and (b) Value loss over training iterations.

RandomAgent is unable to obtain any score, while ScriptAgent benefits from the clear target and therefore wins the highest score 27. Our Agent learns the game strategy from random actions but wastes time on many meaningless steps. It has learned how to build marines to get positive rewards. However, it is stuck by the population limit. It does not know that building the Supply Station can enhance the population limit. So it underperforms compared to ScriptAgent and achieves a score 11.

We try to solve this problem with the improved version, Asynchronous Advantage actor-critic (A3C). We run 8 asynchronous threads at the same time in parallel. Other settings like learning rate and discount ratio are the same to the original actor-critic agent. However, we did not find any obvious advantage of A3C model over the actor-critic agent. It also achieves the most 11 reward and fails to learn building supply to enhance the population limit. In addition, the training process sometimes becomes more unstable. Thus, the A3C agent requires further tuning.

We also record the loss curve of the training process for actor-critic agent as in Figure 5. Specially, policy loss $L_{policy}$ and value loss $L_{value}$ are computed with the following equations.

$$A = v_{target} - v \tag{2}$$
$$L_{policy} = ReduceMean(log(P_a) * A) \tag{3}$$
$$L_{value} = ReduceMean(v * A) \tag{4}$$

where $A$ means the difference between the target value and the predict value and $P_a$ means the probability of taking actions. Figure 5 shows that the loss of our model converges to a low level but the stability needs improvements.

## 5 Conclusion

In this paper, we study the beginning of Starcraft II game. The pipeline for the scripted agent and reinforcement learning-based agents is built on the SC2LE platform. We design both random and scripted agents as a baseline. Our agents are trained with actor-critic and A3C algorithms with detailed analysis. Although their current performance is still inferior to the well-designed scripted agent, it learns much knowledge in this task and is much better than the random agent. This project presents a promising hope for further research in this challenging task.

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
