# OpenReview forum: "Reinforcement Learning for the Beginning of Starcraft II Game"
_CUHK.edu.hk/2021/Course/IERG5350_

### Official Review · AnonReviewer1 · 2020-12-18
**This article uses A3C to solve the beginning resource allocation problem of Starcraft Ⅱ game. Its workload is enough, but there could be other methods to compare different agents.**

**Rating:** 7
**Confidence:** 4

**Review:**

General:

Significance: This article uses Asynchronous Advantage Actor Critic (A3C) to solve the beginning resource allocation problem of StarCraft Ⅱ game, and compares it with  Its work load is enough, but there could be other methods to compare different agents.

Novelty: Use A3C to solve the beginning resource and labor allocation problem of the Real Time Strategy game. The work load is enough for a course project.

Technical quality: This paper uses A3C to solve the focused problem and conduct the experiments on an environment well-defined by DeepMind and Blizzard.

Clarity: The structure of this paper is clear, although sometimes the text ends with unexpected pictures or tables.

Specific:

Pros: a. Define the beginning problem and environment of StarCraft Ⅱ game clearly; b. Use A3C to solve the problem

Cons: a. The final comparison is just the score of different agents who practiced alone in a mini map. Maybe you can try an environment contains two agents on different side of the resource (mine)  with limited amount of the resource. Which could be more practical for a game.

---

### Official Review · AnonReviewer3 · 2020-12-20
**Well played**

**Rating:** 7
**Confidence:** 4

**Review:**

# Summary

This paper applied reinforcement learning methods to the `BuildMarines` task in StarCraft II Learning Environment. The authors implemented Actor Critic and  Asynchronous Advantage Actor Critic agents and compared them with a random agent and a hand-crafted agent. Experiments demonstrate that the trained agents are able to complete the `BuildMarines` task. However, the trained agents are unable to outperform the hand-crafted agent.

# Pros

1. The backgrounds and environment specifications are well-described. The difficulties of training agents in StarCraft II are well-explained. In that case, the task selection is reasonable.
2. The algorithms/agents are well-defined and easy to follow. The design of baseline models is sound and reasonable.

# Cons

1. No external comparison. The AC/A3C agents are only compared with the random agent and the script agent crafted by the authors. In that case, I have no idea whether the performance of agents are good or not for `BuildMarines`. The statistics of `BuildMarines` or the performance of published agents might be necessary to demonstrate how good the mentioned agents are.
2. This paper doesn't explicitly analyze why A3C cannot outperform the AC agent.
3. There are about a dozen typos and grammatical errors in this paper. The citation format and label reference are not correct.
4. Minor: The implemented agents and the training strategy might be limited. Have you tried imitation learning or reward shaping? Partially imitating the scripted agent might lead to interesting results. Besides, it seems that adding a penalty term related to the population could be helpful for the mentioned scenarios.

# Suggestions

1. At first glance, I was unable to spot the difference between the subfigures in Figure 4. The marines might need to be highlighted to demonstrate the difference.
2. Use consistent plotting style in Figure 5 and add `xtitle`/`ytitle`.
3. The source code is suggested to be released for more details.
4. Check the typos in the paper.
5. Choose the correct in-text citation format.